# Integrative Korean Medicine Treatments for Traumatic Facial Palsy Following Mandibular Fracture: A Case Report and Literature Review

**DOI:** 10.3390/healthcare11182546

**Published:** 2023-09-14

**Authors:** Hyunsuk Park, Sook-Hyun Lee, Yeonsun Lee, Dong Joo Choi, Jonghyun Lee, Hyeri Jo, Woojin Jung, Soo-Duk Kim, Hyun A. Sim, Yoon Jae Lee, In-Hyuk Ha

**Affiliations:** 1Department of Korean Medicine Rehabilitation, Bucheon Jaseng Hospital of Korean Medicine, Bucheon 14598, Republic of Korea; drpark1253@jaseng.org (H.P.); cdj0507@jaseng.org (D.J.C.); jongari0810@jaseng.org (J.L.); 2Jaseng Spine and Joint Research Institute, Jaseng Medical Foundation, Gangnam-gu, Seoul 06110, Republic of Korea; sh00god@jaseng.org (S.-H.L.); goodsmile@jaseng.org (Y.J.L.); 3Department of Acupuncture and Moxibustion, Bucheon Jaseng Hospital of Korean Medicine, Bucheon 14598, Republic of Korea; ewidesun@jaseng.org (Y.L.); whgpflgpfl@jaseng.org (H.J.); 4Department of Obstetrics & Gynecology of Korean Medicine, Bucheon Jaseng Hospital of Korean Medicine, Bucheon 14598, Republic of Korea; wj0119@jaseng.org; 5Department of Internal Korean Medicine, Bucheon Jaseng Hospital of Korean Medicine, Bucheon 14598, Republic of Korea; sdhh11@jaseng.org; 6Department of Oriental Neuropsychiatry, Bucheon Jaseng Hospital of Korean Medicine, Bucheon 14598, Republic of Korea; sha0404@jaseng.org

**Keywords:** facial paralysis, mandibular fractures, Korean traditional medicine, case report

## Abstract

Prior studies exploring the effectiveness of traditional Korean medicine (TKM) treatment for facial palsy have mainly focused on Bell’s palsy, and there are few studies on the effectiveness of TKM treatments for traumatic facial palsy following mandibular fracture. The patient was a 24-year-old Korean man with left-sided facial paralysis following a left mandibular fracture. Surgery was performed for the fracture and the facial palsy was treated using conventional medicine (CM) treatments for approximately 3 months, but there was no improvement observed in the patient’s condition. Subsequently, the patient underwent an integrative Korean medicine treatment regimen consisting of acupuncture, pharmacopuncture, cupping, moxibustion, and herbal medication for a duration of 2 months. After 2 months of treatments, the House–Brackmann facial grading scale changed from Ⅴ to II and Yanagihara’s unweighted grading score increased from 9 to 34. This case presentation and previous studies of traumatic facial palsy using TKM treatment show that TKM treatment may be considered a complementary or alternative treatment method to CM treatment in patients with traumatic facial palsy. PROSPERO registration number: CRD42023445051.

## 1. Introduction

The facial nerve travels long courses compared to those of other nerves and is distributed over a wide region of the face, which makes it vulnerable to head and face trauma due to road accidents, falls, etc. [1,2]. In addition, since the nerve distribution pattern varies for each individual, it is difficult to predict nerve travel during head and face surgery, which increases the likelihood of surgery-induced traumatic facial nerve injury [3,4].

Traumatic facial palsy is associated with a higher rate of nerve damage, resulting in slower recovery and a poorer prognosis than those for Bell’s palsy; thus, it is imperative that the patient receives optimal treatment at the earliest [4]. In CM, the general practice is to perform surgery as soon as possible in the case of trauma-induced complete facial palsy, whereas steroids and antivirals are used for the conservative treatment of incomplete facial palsy. Surgical intervention is considered if symptoms do not improve after four months of conservative management [5].

However, when the side effects of conventional medical drugs and surgical treatments are not small, TKM treatments can be used as an alternative treatment method [6,7]. TKM treatments for facial palsy include acupuncture, herbal medicine, pharmacopuncture, and the embedded needle (Maesun) technique. Li et al. reported that acupuncture treatment is effective for peripheral facial palsy [3,8]. However, previous studies investigating the effectiveness of TKM treatment for facial palsy have mainly focused on Bell’s palsy, an idiopathic facial palsy [3,5,9], indicating that there are few studies on the effectiveness of TKM treatments for traumatic facial palsy.

We present a case report on the treatment outcomes of a patient who (1) had traumatic facial palsy following mandibular fracture and underwent CM treatment for approximately three months with no improvement; (2) visited the Facial Palsy Clinic of Bucheon Jaseng Hospital of Korean Medicine and underwent integrative TKM treatments with remarkable improvement. Furthermore, we present the results of a systematic review of publications and data on the treatment of traumatic facial palsy using TKM to establish reference data that may be useful to evaluate the effect of TKM treatments on traumatic facial palsy and the associated prognosis.

## 2. Detailed Case Description

We present a retrospective case report of a patient with traumatic facial palsy caused by mandibular fracture, who underwent treatment at Jaseng Hospital of Korean Medicine for two months, from 3 November 2022 to 2 January 2023. This study was reviewed and approved by the Institutional Review Board of the Jaseng Hospital of Korean Medicine (JASENG 2023-01-002). The patient agreed to participate in the study and gave informed consent. The treatment and assessment presented in the following section were conducted by two specialist Korean medicine doctors (KMDs).

### 2.1. Patient Information

The patient was a 24-year-old Korean man with no family or social history. On 8 August 2022, the patient fell from the second floor while playing with an acquaintance, and suffered traumatic facial palsy following left mandibular fracture. He underwent surgery for the mandibular fracture, followed by outpatient care at another hospital for approximately three months to treat facial palsy; however, no improvement was observed. Thus, the patient visited Bucheon Jaseng Hospital of Korean Medicine on 3 November 2022.

Electromyography results at another hospital showed “no response from the left facial nerve”, and ear, nose, and throat test results reported normal tympanometry and audiometry findings. No abnormal findings were reported via brain computed tomography performed on 9 August 2022 and 1 September 2022 at U Hospital and on 19 October 2022 at I Hospital.

At the patient’s first visit to Bucheon Jaseng Hospital of Korean Medicine, no movement was observed on the left forehead; thus, no wrinkles were seen on the left forehead, and when the patient tried to close his eyes, the incomplete closure led to >5 mm corneal exposure.

### 2.2. Treatments

The treatments administered in this case were acupuncture, pharmacopuncture, moxibustion, cupping, herbal medicine, and Chuna manual therapy.

In total, 29 sessions of acupuncture treatment were performed. The frequency of acupuncture treatment was three to four weekly sessions for the first month, followed by two to three sessions weekly. Standardized disposable stainless-steel filiform needles (0.25 mm × 30 mm, Dongbang Acupuncture Inc., Seongnam-si, Gyeonggi-do, Republic of Korea) were used to administer the acupuncture treatment. The needles were inserted and retained for 15 min at acupoints TE17 (Yifeng), LI4 (Hegu), ST4 (Dicang), ST6 (Jiache), GB20 (Fengchi), ST3 (Juliao), SI18 (Quanliao), LI20 (Yingxiang), BL2 (Zanzhu), and TE23 (Sizhukong) of the affected side. Moreover, electroacupuncture was performed at a frequency of 3 Hz over acupoints ST4 (Dicang) to ST6 (Jiache) and BL2 (Zanzhu) to TE23 (Sizhukong) of the affected side.

In total, 29 sessions of pharmacopuncture treatment were performed. Hominis Placental pharmacopuncture solution (Jaseng Herbal Dispensary) and Hwangreonhaedok-tang pharmacopuncture solution (Jaseng Herbal Dispensary) were injected (0.5 cc each) into acupoints GB20 (Fengchi), TE17 (Yifeng), ST7 (Xiaguan), and EX-HN5 (Taiyang) of the affected side. Regarding the pharmacopuncture solution, the patient underwent 22 sessions of Hominis Placenta pharmacopuncture and seven sessions of Hwangreonhaedok-tang pharmacopuncture.

In total, 29 sessions of moxibustion treatment were administered. During each visit, electronic moxibustion (On Tteum, HS Medix, 33 mm × 20 mm) was administered to some sites, including GB12 (Wangu) and GB20 (Fengchi). In total, 29 sessions of cupping treatment were administered using a disposable cup (Dongbang Medical, 28 mm) for dry or wet cupping at sites GB20 (Fengchi) or TE17 (Yifeng).

As for herbal medicine treatment, Wasahaepyo-tang was prescribed twice in 2-week intervals. The first round of prescription was from 16 November 2022 to 30 November 2022, during which the patient took one pack (100 cc) twice daily, morning and evening. The second round of prescription was from 2 December 2022 to 16 December 2022, during which the patient took one pack (100 cc) twice daily, morning and evening. The composition and amount of medicinal herbs in Wasahaepyo-tang are presented in Table 1.

As for Chuna manual therapy, a manipulation technique, the SJS non-resistance technique-facial palsy (SJSNRT-F), was applied five times, on 5 November, 8 November, 10 November, 11 November and 16 November 2022. The therapist placed both hands on the patient’s face and applied force to pull the skin of the affected side for 10 s, and gently and slowly pushed the skin of the unaffected side for 10 s. The force applied when pulling the skin was such that the therapist was able to sense the skin moving while the applied pressure was barely detectable on the muscles underneath the skin (Figure 1) [10]. Considering this described procedure as one set, four sets were administered and the therapist checked the alignment of the forehead, philtrum, and middle of the lips in terms of symmetry and straight-line alignment. After completing each SJSNRT-F session, the patient was instructed to rest for at least 1 h without using the facial muscles [10,11].

### 2.3. Outcome Measures

The House–Brackmann facial grading scale (HB Grade) and Yanagihara’s unweighted grading score (Yanagihara’s point) were used to assess the patient’s symptoms. The HB grade is used to assess facial palsy by quantitatively characterizing the facial nerve dysfunction and neurological sequelae of facial nerve damage. Based on the severity of facial paralysis, the HB grade classifies facial palsy into six grades (grade I to grade Ⅵ), and a higher score indicates more severe paralysis [12]. Yanagihara’s point allows the assessment and grading of the movement of facial muscles. The system measures 10 separate functions of different facial muscles by scoring the severity of dysfunction into 5 grades, from 0 (complete palsy) to 4 (nearly normal) [13].

### 2.4. Follow-Up and Outcome

On 3 November 2022, the patient visited Bucheon Jaseng Hospital of Korean Medicine for the first time with the chief complaint of left-sided facial palsy following mandibular fracture three months prior to the time of visit. During this visit, no movement of muscles on the left forehead was observed; thus, no wrinkles were seen on the left forehead, and when the patient tried to close his eyes, an incomplete closure of the eyelids resulted in a corneal exposure of >5 mm. In addition, when uttering “i” and “oh” sounds, the patient was unable to move his lips to make the appropriate shapes. He also reported abnormalities in his sense of taste. No abnormalities in audiometry and facial sensation examination were observed, and the patient did not complain of postauricular pain. During the visit, assessment results showed HB grade V and Yanagihara’s point 9.

On 10 November 2022 (week 2 of treatment), movement of some muscles was observed in the zygomatic region, and the patient also reported that he sensed small movement in the area. The assessment results were HB grade IV and Yanagihara’s point 15. The administered TKM treatments were acupuncture, pharmacopuncture, moxibustion, cupping, and Chuna manual therapy.

On 16 November 2022 (week 3), a slight improvement was observed when closing the eyelid along with some improvement in the movement of muscles in the zygomatic region. The assessment results were HB grade IV and Yanagihara’s point 18. The administered TKM treatments were acupuncture, pharmacopuncture, moxibustion, and cupping; Wasahaepyo-tang was prescribed for 15 days as a herbal medicine treatment.

On 24 November 2022 (week 4), improvement was observed in the overall facial movements of the patient, and the patient also reported improvement in eyelid closure. The assessment results were HB grade III and Yanagihara’s point 24. The administered TKM treatments were acupuncture, pharmacopuncture, moxibustion, and cupping, along with the herbal medicine, Wasahaepyo-tang.

On 2 December 2022 (week 5), no remarkable differences compared to the outcomes of week 4 were observed; however; a slight improvement in the movement of the periocular region was observed. The assessment results were HB grade III and Yanagihara’s point 24. The administered TKM treatments were acupuncture, pharmacopuncture, moxibustion, and cupping; Wasahaepyo-tang was prescribed for an additional 15 days as a herbal medicine treatment.

On 8 December 2022 (week 6), the patient showed more improvement in the movement of both periocular and perioral regions. However, the improvement in the movement of the perioral region was observed to be slower than that of the periocular region. The assessment results were HB grade II and Yanagihara’s point 30. The administered TKM treatments were acupuncture, pharmacopuncture, moxibustion, and cupping, along with the herbal medicine, Wasahaepyo-tang.

On 17 December 2022 (week 7), forehead wrinkling movement remained limited when the patient tried to open his eyes wide; however, when relaxed, there was no remarkable difference between the affected and unaffected side. Movement in the perioral region also improved significantly compared to that at week 6, resulting in ease of mouth movement when eating. The assessment results were HB grade II and Yanagihara’s point 30. The administered TKM treatments were acupuncture, pharmacopuncture, moxibustion and cupping.

On 28 December 2022 (week 9), the movement in the perioral region showed an overall improvement. For movements when enunciating “oh” and “i” sounds, the patient’s functions were restored almost to the level of unaffected people. The assessment results were HB grade II and Yanagihara’s point 34. The administered TKM treatments were acupuncture, pharmacopuncture, moxibustion, and cupping.

The timeline of visits and treatments are presented in Figure 2 and Figure 3, and Table 2.

### 2.5. Literature Review

Following a literature search of eight domestic and foreign databases in June 2023 to analyze the research trend of integrative TKM for traumatic facial palsy, 25 case reports were identified (Appendix A). The full search terms, strategies and flow chart are listed in Appendix A.

## 3. Discussion

Traumatic facial palsy is associated with a higher rate of nerve damage and poorer prognosis than are other types of facial palsy [4]. In modern society, as a consequence various socioeconomical changes, such as an increasing incidence of road traffic accidents due to more frequent travels and the faster development of transportation, and increasing plastic surgery and corrective jaw surgeries (orthognathic surgery) due to the advancement of healthcare technology and rising interest in the beauty industry, there is an increasing incidence of post-traumatic facial paralysis [3].

In the present study, we present a case report of a patient with traumatic facial palsy caused by mandibular fracture, who experienced no improvement after CM treatment over three months before the visit to the Facial Palsy Clinic of Bucheon Jaseng Hospital of Korean Medicine, but showed a noticeable improvement in treatment outcomes after undergoing integrative TKM treatment sessions, consisting of acupuncture, electroacupuncture, cupping, Chuna manual therapy, and the consumption of Wasahaepyo-tang.

Wasahaeepyo-tang includes medicinal herbs such as Zelan (*Lycopus lucidus*), Shoudihuang (*Rehmania glutinosa*), Renshen (*Panax ginseng*), Chenpi (*Citrus unshiu*), and Shengjiang (*Zingiber officinale*) (Table 1). Zelan shows effective neuroprotection via inhibiting the activation of NOD-like the receptor family, Pyrin domain-containing 3 (NLRP3) and inflammasome (a multiprotein complex initiating an inflammatory form of cell death), and the effects of nerve regeneration via increasing the expression of brain-derived neurotrophic factor (BDNF) and nerve growth factor [14]. Catapol, a component of Shoudihuang, inhibits apoptosis and increases BDNF expression [15,16]. Renshen functions in various pathways that promote BDNF expression to provide nerve regeneration and neuroprotective effects [17]. Chenpi has anti-inflammatory effects, thus reducing acute inflammation [11]. Shengjiang has been linked with Cyclooxygenase-2 inhibition to reduce Prostaglandin E2 levels and exhibits analgesic and anti-inflammatory effects [18]. Considering the effectiveness of the main medicinal herbs in Wasahaepyo-tang, it is believed that Wasahaepyo-tang makes a considerable contribution to the improvement of facial palsy symptoms [14,15,16,17,18].

SJSNRT-F is a type of Chuna manual therapy for the treatment of facial palsy, and is known to restore asymmetrical facial muscles to their original position and stimulate proprioceptive neuromuscular receptors [10,11,19]. Prior studies on SJSNRT-F include three case reports by Lee et al. [10], Choi et al. [11], and Kyung et al. [19]. In the present case presentation, the patient refused Chuna manual therapy in the middle of the treatment schedule because of cost burdens. We believe that a faster and more complete recovery from facial palsy would have been achieved if Chuna manual therapy was continued. In addition, future well-designed studies examining and verify the effect of Chuna manual therapy on patients with facial palsy are necessary.

An analysis of the interventions in previous studies on traumatic facial palsy using TKM treatment revealed that all the studies (100%) used acupuncture and eight studies (32.0%) used herbal medicine treatment. All the studies used herbal medicine treatment as an intervention applied in a combined therapy of acupuncture and herbal medicine treatment. In addition to acupuncture and herbal medicine treatment, six studies (24.0%) used integrative TKM treatment, including moxibustion, cupping or pharmacopuncture, and Chuna manual therapy, as an intervention. In the seven studies that used integrative TKM treatment for facial palsy, a combination of acupuncture and herbal medicine was used, whereas pharmacopuncture was used in two studies, moxibustion in three studies, and cupping in two studies. When administering TKM treatment, the treatment effect is thought to be synergistic if pharmacopuncture, moxibustion, and cupping therapy are combined with acupuncture and herbal medicine.

There are different causes of traumatic facial palsy, including skull fracture, temporal bone fracture, and complications of facial surgery. In the literature reviewed in this study, there are reports on the effect of TKM treatment on traumatic facial palsy caused by various types of trauma. However, there are no studies on traumatic facial palsy following mandibular fracture. Thus, this case report is of great significance because it is the first case report with TKM treatment for traumatic facial palsy caused by mandibular fracture.

An analysis of the literature selected in this study revealed that the outcome measures used in the literature showed that 16 (64.0%), 5 (20.0%), and 4 (16.0%) studies used the total effective rate (TER), HB grade, and Yanagihara’s point, respectively, as the outcome measure. Currently, the most commonly recognized outcome measures for the assessment of facial palsy are the HB grade and Yanagihara’s point [12,20]. Regarding TER, the criteria for grading the degree of improvement varies in the literature, making an objective determination of the degree of improvement difficult. Thus, future studies need to actively utilize internationally recognized outcome measures, such as the HB grade or Yanagihara’s point.

Although the optimal timing of surgical intervention for patients with traumatic facial palsy is not clearly defined, Xie et al. reported that surgery may be beneficial if performed within two months; however, a complete recovery rate of 16% has been reported in patients who underwent surgery after two months from the onset of the condition, showing a significantly worse prognosis than that of patients who underwent surgery within two months of onset [21]. The condition of patients with traumatic facial palsy is poor because of head and face trauma, and interventions to determine the survival of the patient are required in many cases, resulting in delays in diagnosis and treatment [22]. In particular, if a patient who has had traumatic facial palsy for more than two months visits the hospital, selecting conservative management with TKM treatment and monitoring the responses and course of the condition may be an alternative approach to avoiding surgical intervention.

In the present case, the patient and research team initially expected a poor prognosis. However, this case report demonstrates the significance of integrative TKM treatment in that the patient unresponsive to CM treatment achieved good prognosis and fast recovery after the TKM treatment. We believe that an even better outcome could have been attained if the patient had actively received TKM treatment from the onset of the facial palsy. It has been generally recognized that improvement of patients with post-traumatic facial palsy through TKM treatment is more difficult to achieve than is that of patients with Bell’s palsy [4]. However, based on the findings of this case report and prior studies on traumatic facial palsy using TKM treatment, integrative TKM treatment may be considered an alternative for the treatment of traumatic facial palsy.

Our study had several limitations. This case report is only on a single case, and the number of cases is too small to use the findings of this case report as evidence for verifying the effect of TKM treatment. Moreover, since multiple interventions were used in combination, it is difficult to evaluate or determine the effects of individual interventions. Furthermore, this cases study is deficient in objective visual data for understanding the patient’s progress. Therefore, further studies to address and improve these limitations need to be conducted. Furthermore, we believe that high-quality studies are warranted in this area.

## 4. Conclusions

This case report confirms that after administering integrative TKM treatment to a patient with traumatic facial palsy caused by mandibular fracture, the HB grade improved from grade V to grade II, and Yanagihara’s point improved from 9 points to 34 points, indicating that integrative TKM treatment led to remarkable recovery, compared to the patient’s condition at the baseline. Additionally, the literature review in this study revealed that TKM treatment is generally effective for traumatic facial palsy and improves symptoms to some extent.

Based on the findings of this case report and prior studies on traumatic facial palsy using TKM treatment, integrative TKM may be considered an alternative for the treatment of traumatic facial palsy.

## Figures and Tables

**Figure 1 healthcare-11-02546-f001:**
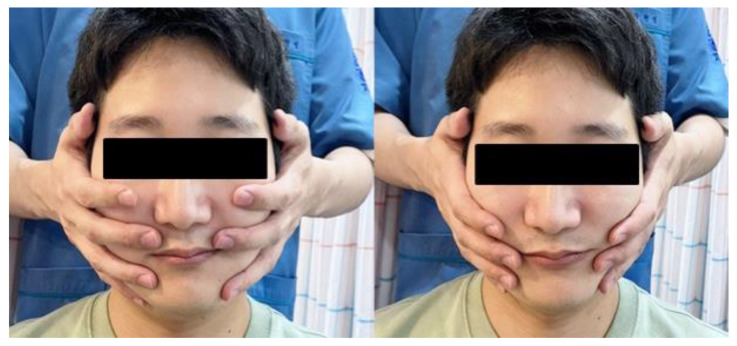
SJS non-resistance technique-facial palsy (SJSNRT-F) for the treatment of facial nerve palsy.

**Figure 2 healthcare-11-02546-f002:**
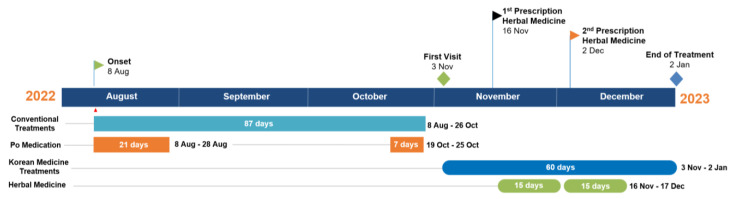
Timeline of CM and TKM treatments received by the patient for 5 months.

**Figure 3 healthcare-11-02546-f003:**
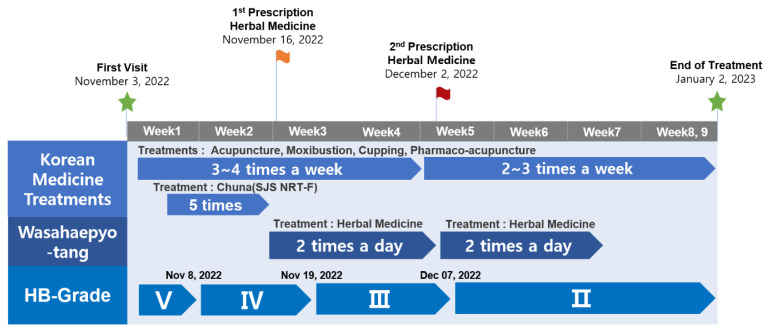
Timeline of integrative TKM treatment and the changes in the HB Grade.

**Table 1 healthcare-11-02546-t001:** Composition and quantity of medicinal herbs in Wasahaepyo-tang.

Herbal Name	Botanical Name	Medicinal Part	Dosage (g/Po)
甘草 (Gancao)	*Glycyrrhiza uralens*	Root, Rhizome	2.5
當歸 (Danggui)	*Angelica gigas*	Root	2.5
山藥 (Shanyao)	*Dioscorea batatas*	Rhizome	5
生薑 (Shengjiang)	*Zingiber officinale*	Rhizome	12.5
熟地黃 (Shoudihuang)	*Rehmania glutinosa*	Root	7.5
升麻 (Shengma)	*Cimicifuga heracleifolia*	Rhizome	0.75
柴胡 (Chaihu)	*Bupleurum falcatum*	Root	0.75
人蔘 (Renshen)	*Panax ginseng*	Root	5
陳皮 (Chenpi)	*Citrus unshiu*	Pericarp	2.5
澤蘭 (Zelan)	*Lycopus lucidus*	Above ground	2.5

**Table 2 healthcare-11-02546-t002:** Changes in Yanagihara’s Point after integrative TKM treatment.

Yanagihara’s Point	First Visit	Week 2	Week 3	Week 4	Week 5	Week 6	Week 7	Week 9
At rest	2	2	2	2	2	3	3	4
Wrinkled forehead	0	0	1	2	2	3	3	3
Blinking	0	1	2	2	2	3	3	3
Light closure of eye	0	1	2	2	2	3	3	3
Tight closure of eye	1	2	2	3	3	3	3	4
Closure of eye on affected side only	1	2	2	2	2	3	3	3
Wrinkled nose	1	2	2	3	3	3	3	3
Whistling	2	2	2	3	3	3	3	4
Grinning	1	2	2	3	3	3	3	4
Depressed lower lip	1	1	1	2	2	3	3	3
Total Score	9	15	18	24	24	30	30	34

## Data Availability

The data presented in this study are available on request from the corresponding author. The data are not publicly available due to privacy/ethical restrictions.

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
