# Peer review of "Integrative Korean Medicine Treatments for Traumatic Facial Palsy Following Mandibular Fracture: A Case Report and Literature Review"

_healthcare, 2023, doi:10.3390/healthcare11182546_

Round 1

Reviewer 1 Report

Dear authors, a manuscript is provided that combines two methodologies, a case report and a literature review.

To report a clinical case report you should use the CARE guidelines; while for a review you should use PRISMA (it is not clear for me that it is a systematic review, but it has been published in PROSPERO and therefore meets the criteria from this point of view).

From the perspective of a case report including a review of the literature, the title is correct.

Keywords: Review and adjust keywords to MeSH terms.

Abstract: Following CARE guidelines is correct. The authors provide the PROSPERO registration number.

Introduction: correct,.

Case description: Adequate. Approval of the ethics committee reported. Must add that the patient has given informed consent to provide personal data and photos.

Clinical finds, timeline, diagnostic assessment, therapeutic intervention, follow-up and outcomes reported.

Discussion includes review of the literature. It is not incorrect what the authors have done but, in my opinion, it is more correct to expose the results of the review in a section after the presentation of the case and before the discussion (in the discussion these results will be discussed). In this way, the methodology followed in the review and the results achieved can be presented in greater detail. A flow chart with the records retrieved in the search, duplicates eliminated and screening (PRISMA criteria) should be added in a supplementary file.

The CARE guidelines recommend including the patient's perspective in the clinical case. A sentence can be added in the discussion in this regard.

References should be reviewed according to Healthcare journal guidelines. Include link to the primary source (preferably the DOI when available).

Author Response

[Reviewer1]

Dear Reviewer 1,

We did our best to reflect your opinions as much as possible and to raise our paper quality to suitable for publishing at Healthcare. And we are very delighted an grateful for your kind comments. We appreciate again your further comments and have striven to incorporate them in our revised manuscript again. In the following sections, please find our responses to each of your comments and suggestions.

Dear authors, a manuscript is provided that combines two methodologies, a case report and a literature review.

To report a clinical case report you should use the CARE guidelines; while for a review you should use PRISMA (it is not clear for me that it is a systematic review, but it has been published in PROSPERO and therefore meets the criteria from this point of view).

From the perspective of a case report including a review of the literature, the title is correct.

Keywords: Review and adjust keywords to MeSH terms.

  • Response: Thank you for your kind comment. We checked our manuscript again, and fixed the keywords as follows:

(Line 36) Keywords: Facial Paralysis; Mandibular fractures; Korean Traditional Medicine; Case reports

Abstract: Following CARE guidelines is correct. The authors provide the PROSPERO registration number.

Introduction: correct,.

Case description: Adequate. Approval of the ethics committee reported. Must add that the patient has given informed consent to provide personal data and photos.

  • Response: Thank you for your kind comment. We checked our manuscript again, and added the sentences as follows:

(Line 79) The patient agreed to participate in the study and gave informed consent. We submitted the patient consent form.

Clinical finds, timeline, diagnostic assessment, therapeutic intervention, follow-up and outcomes reported.

Discussion includes review of the literature. It is not incorrect what the authors have done but, in my opinion, it is more correct to expose the results of the review in a section after the presentation of the case and before the discussion (in the discussion these results will be discussed). In this way, the methodology followed in the review and the results achieved can be presented in greater detail. A flow chart with the records retrieved in the search, duplicates eliminated and screening (PRISMA criteria) should be added in a supplementary file.

  • Response: Thank you for your kind comment. We checked our manuscript again, and present the results of the review in '2.5. Literature review'. Additionally, We include a flow chart in the Supplementary table.

(Line 210) 2.5. Literature Review

Following a literature search of eight domestic and foreign databases in June 2023 to analyze the research trend of integrative TKM for traumatic facial palsy, 25 case reports, 1 chart review study, and 4 randomized controlled trials (RCTs) were identified (Table 3). The full search terms, strategies and flow chart are listed in Supplementary Table.

The CARE guidelines recommend including the patient's perspective in the clinical case. A sentence can be added in the discussion in this regard.

  • Response: Thank you for your kind comment. We checked our manuscript again, and added the sentences as follows:

(Line 304) The patient in this case and research team believe that an even better outcome could have been attained if the patient had actively received TKM treatment from the onset of the facial palsy.

References should be reviewed according to Healthcare journal guidelines. Include link to the primary source (preferably the DOI when available).

  • Response: Thank you for your kind comment. We have rechecked our manuscript, and corrected the references.

For more details, please see the revised manuscript.

Reviewer 2 Report

The first time you include acronyms within the text, you have to write them in full. After that, you should report them as abbreviations only.

In the case report, conclusion (This case and previous studies on traumatic facial palsy using TKM suggest that TKM 32 should be considered a complementary and alternative treatment method to conventional medicine. 33 For patients with traumatic facial palsy caused by mandibular fracture for >2 months, combining 34 conservative treatment with TKM and observing the response and progression may be an alternative to surgical intervention.) is not logical, should be interpreted with caution

Check the keywords based on the mesh.

Line 45-62 one reference is used, which is not appropriate.

Line 75 needs a reference.

The titles of figures and tables should be clear.

In Discussion, all sentences need references, also it's too long.

Conclusions is long, it should be summarized and please only present key findings

The article should be revised based on the The CAREGuidelines: Consensus-based Clinical Case Reporting Guideline Development.

Just considering the type of article, it is a little long, so it should be more concise in all sections.

Author Response

[Reviewer 2]

Dear Reviewer 2,

We did our best to reflect your opinions as much as possible and to raise our paper quality to suitable for publishing at Healthcare. And we are very delighted an grateful for your kind comments. We appreciate again your further comments and have striven to incorporate them in our revised manuscript again. In the following sections, please find our responses to each of your comments and suggestions.

The first time you include acronyms within the text, you have to write them in full. After that, you should report them as abbreviations only.

  • Response: We checked our manuscript again, and corrected the some words to abbreviations (Line 52),(Line 59).

In the case report, conclusion (This case and previous studies on traumatic facial palsy using TKM suggest that TKM 32 should be considered a complementary and alternative treatment method to conventional medicine. 33 For patients with traumatic facial palsy caused by mandibular fracture for >2 months, combining 34 conservative treatment with TKM and observing the response and progression may be an alternative to surgical intervention.) is not logical, should be interpreted with caution.

  • Response: We checked our manuscript again, and found our mistake. We corrected the sentences as follows:

(Line 31) This case presentation and previous studies of traumatic facial palsy using Traditional Korean medicine (TKM) treatment show that TKM treatment may be considered a complementary or alternative treatment method to CM treatment in patients with traumatic facial palsy.

Check the keywords based on the mesh.

  • Response: We checked our manuscript again, and fixed the keywords as follows:

(Line 36) Keywords: Facial Paralysis; Mandibular fractures; Korean Traditional Medicine; Case reports

Line 45-62 one reference is used, which is not appropriate.

  • Response: We checked our manuscript again and deleted the sentence during the revision.

Line 75 needs a reference.

  • Response: We checked our manuscript again, and updated the references as follows.
  1. Kim, M.J.; Song, J.Y.; Song, W.S.; Kim, P.K.; Ryu, H.K.; Park, Y.C., et al. Clinical Study on Peripheral Facial Nerve Injury. J. Acupunct Res. 2012, 29, 23-34, doi:10.13045/kamms.2012083.
  2. Ahn, H.L.; Shin, M.S. A Case Report of a Patient with Facial Nerve Paralysis Caused by Traumatic Temporal Bone Fracture. J. Korean Med Rehabi. 2007, 17, 159-166.
  3. Lee, J.M.; Kim, E.M.; Song, H.G.; Go, S.K.; Kim, S.L.; Kim, J.H., et al. Clinical Study of Two Patients with Deviation of the Eye and Mouth Caused by Trauma. J. Acupunct Res. 2006, 23, 81-89.

The titles of figures and tables should be clear.

  • Response: We checked our manuscript again, and fixed the titles of figures and tables

(Line 206) Timeline of CM and TKM treatments received by the patient for 5 months

(Line 208) Timeline of integrative TKM treatment and the changes in the HB Grade

(Line 209) Changes in Yanagihara’s Point after integrative TKM treatment

In Discussion, all sentences need references, also it's too long.

  • Response: We checked our manuscript again, and rewrite the Discussion. We kindly ask you to check again.

Conclusions is long, it should be summarized and please only present key findings

  • Response: We checked our manuscript again, and rewrite the Conclusions. We kindly ask you to check again.

(Line 318) This case report confirms that after administering integrative TKM treatment to a patient with traumatic facial palsy caused by mandibular fracture, the HB grade improved from grade V to grade II, and Yanagihara's point improved from 9 points to 34 points, indicating that integrative TKM treatment achieved remarkable recovery, compared to the patient’s condition at baseline. And the literature review in this study revealed that TKM treatment is generally effective for traumatic facial palsy and improves symptoms to some extent.

Based on the findings of this case report and prior studies on traumatic facial palsy using TKM treatment, integrative TKM may be considered as an alternative for the treatment of traumatic facial palsy.

The article should be revised based on the The CAREGuidelines: Consensus-based Clinical Case Reporting Guideline Development.

  • Response: We checked the CAREGuidelines and our manuscript again. We added several missing part(Patient Perspective, Informed Consent) as follows:

Patient Perspective

(Line 304) The patient in this case and research team believe that an even better outcome could have been attained if the patient had actively received TKM treatment from the onset of the facial palsy

Informed Consent

(Line 79) The patient agreed to participate in the study and gave informed consent.

Just considering the type of article, it is a little long, so it should be more concise in all sections.

  • Response: We checked our manuscript again, and corrected the manuscript to be more concise. We kindly ask you to check again.

For more details, please see the revised manuscript.

Reviewer 3 Report

1. The abstracts lacks of focusing the aim and objective of the study

2. Please rephrase the line 29

3. Introduction sounds reflexive and adequate but the setbacks of steroid drugs are not addressed because limitations of synthetic drugs reinforce us to use herbal/alternative drugs

4. Table 1 needs more information, which parts of the plants have been used for the herbal treatment?

5. As the alternative treatments are applied concomitantly, how can you evaluate the effect of individual treatment? Which one was the best in resolving the problem?

6. The Yanagihara's scores are important to be compared with a control to understand the improvement

7. Discussion is too long, results repeated in this part. Authors should revise the discussion to extensively summarize it and to avoid the repetition of results.

8. What about the drawbacks of the study? How can those be resolved?

Author Response

[Reviewer 3]

Dear Reviewer 3,

We did our best to reflect your opinions as much as possible and to raise our paper quality to suitable for publishing at Healthcare. And we are very delighted an grateful for your kind comments. We appreciate again your further comments and have striven to incorporate them in our revised manuscript again. In the following sections, please find our responses to each of your comments and suggestions.

  1. The abstracts lacks of focusing the aim and objective of the study
  • Response: We checked our manuscript again, and rewrite the abstract as follows:

(Line 22) Prior studies exploring the effectiveness of TKM treatment for facial palsy have mainly focused on Bell's palsy, and there are few studies on the effectiveness of TKM treatments for traumatic facial palsy following mandibular fracture. The patient was a 24-year-old Korean man with left-sided facial paralysis following a left mandibular fracture. Surgery was performed for the fracture and the facial palsy was treated using conventional medicine (CM) treatments for approximately 3 months, but there was no improvement observed in the patient's condition. Subsequently, the patient underwent a integrative Korean medicine treatment regimen consisting of acupuncture, pharmacopuncture, cupping, moxibustion, and herbal medication for a duration of 2 months. After 2 months of treatments, House-Brackmann facial grading scale changed from â…¤ to II and Yanagihara’s unweighted grading score increased from 9 to 34. This case presentation and previous studies of traumatic facial palsy using Traditional Korean medicine (TKM) treatment show that TKM treatment may be considered a complementary or alternative treatment method to CM treatment in patients with traumatic facial palsy.

  1. Please rephrase the line 29
  • Response: We checked our manuscript again, and fixed the sentences as follows:

(Line 27,28) Subsequently, the patient underwent a integrative Korean medicine treatment regimen consisting of acupuncture, pharmacopuncture, cupping, moxibustion, and herbal medication for a duration of 2 months. After 2 months of treatments, House-Brackmann facial grading scale changed from â…¤ to II and Yanagihara’s unweighted grading score increased from 9 to 34.

  1. Introduction sounds reflexive and adequate but the setbacks of steroid drugs are not addressed because limitations of synthetic drugs reinforce us to use herbal/alternative drugs
  • Response: We checked our manuscript again, and added the sentences as follows:

(Line 57) However, in when where the side effects of conventional medical drugs and surgical treatments are not small, TKM treatments can be used as an alternative treatment method

  1. Table 1 needs more information, which parts of the plants have been used for the herbal treatment?
  • Response: We added medicinal part in table 1(line 129,130). We kindly ask you to check again.
  1. As the alternative treatments are applied concomitantly, how can you evaluate the effect of individual treatment? Which one was the best in resolving the problem?
  • Response: What you pointed out is one of the limitations of this study. Referring to the literature review we conducted, we were able to confirm that acupuncture treatment is mainly used for patients with traumatic facial palsy. However, We believe that high-quality studies should be actively conducted to confirm the effectiveness of individual interventions.

(Line 254) Analysis of the interventions in previous studies on traumatic facial palsy using TKM treatment revealed that 29 studies (96.7%) used acupuncture

(Line 313 ) Moreover, since multiple interventions were used in combination, it is difficult to evaluate or determine the effects of individual interventions.

  1. The Yanagihara's scores are important to be compared with a control to understand the improvement
  • Response: We kindly request you to consider that our paper is a case report and, as such, generally does not include a control group.
  1. Discussion is too long, results repeated in this part. Authors should revise the discussion to extensively summarize it and to avoid the repetition of results.
  • Response: We checked our manuscript again, and fixed the manuscript to be more concise. We kindly ask you to check again.
  1. What about the drawbacks of the study? How can those be resolved?
  • Response: We checked our manuscript again, and presented the limitations in discussion

(Line 311) Our study had several limitations. This case report is only a single case, and the number of cases is too small to use the findings of this case report as evidence of verifying the effect of TKM treatment. Moreover, since multiple interventions were used in combination, it is difficult to evaluate or determine the effects of individual interventions. Furthermore, this cases study is deficient in objective visual data to understand the patient's progress. Therefore, further studies to address and improve these limitations need to be conducted. Furthermore, we believe that high-quality studies are warranted in this area.

For more details, please see the revised manuscript.

Reviewer 4 Report

I congratulate the authors on their efforts.

It is important in a scientific reading what kind of work it focuses on.

In this case it shows scientific evidence with a treatment programme, this paper is more developed as a review and I encourage the authors to focus it in my view in a systematic review and meta-analysis.

Thanks for reading

Additional comments:

The article "Clinical case and literature review" does not present a correct methodology, because from my point of view and more focused on traditional medicine, it needs scientific evidence to prove its efficacy, i.e. to present it as a clinical trial or as a systematic review-meta-analysis.

Author Response

[Reviewer 4]

Dear Reviewer 4,

We did our best to reflect your opinions as much as possible and to raise our paper quality to suitable for publishing at Healthcare. And we are very delighted an grateful for your kind comments. We appreciate again your further comments and have striven to incorporate them in our revised manuscript again. In the following sections, please find our responses to each of your comments and suggestions.

I congratulate the authors on their efforts.

It is important in a scientific reading what kind of work it focuses on.

In this case it shows scientific evidence with a treatment programme, this paper is more developed as a review and I encourage the authors to focus it in my view in a systematic review and meta-analysis.

Thanks for reading

Additional comments:

The article "Clinical case and literature review" does not present a correct methodology, because from my point of view and more focused on traditional medicine, it needs scientific evidence to prove its efficacy, i.e. to present it as a clinical trial or as a systematic review-meta-analysis.

  • Response: First of all, thank you for your kind comment. The method we applied to the paper was in the form of a 'case report and literature review'. This format is often used in the fields of medicine and medical research, reporting new cases and reviewing related literature to contribute to expanding and sharing knowledge. We understand that in terms of scientific evidence, it would be more appropriate to write in a 'systematic review' format. However, the topic we study has a limited number of relevant RCTs in the literature. Therefore, we took due care of this limited information, and decided that writing the paper in the form of a ‘case report and literature review’ would be more appropriate in this case.

We also addressed about it in discussion.

Line (266) : Considering the number of published studies on traumatic facial palsy, the number of RCTs and literature reviews that report the effectiveness of TKM treatment for traumatic facial palsy may be smaller than that on idiopathic facial palsy (Bell's palsy). Thus, future studies with high level of evidence on the effect of TKM treatment on traumatic facial palsy are warranted.

Please understand that we did not choose a ‘systematic review’ due to limited resources.

For more details, please see the revised manuscript.

Round 2

Reviewer 2 Report

According to the type of article (case report), if possible, it should be more concise.

Author Response

Dear Reviewer 2,

We did our best to reflect your opinions as much as possible and to raise our paper quality to suitable for publishing at Healthcare. And we are very delighted an grateful for your kind comments. We appreciate again your further comments and have striven to incorporate them in our revised manuscript again. In the following sections, please find our responses to each of your comments and suggestions.

Reviewer's comment: According to the type of article (case report), if possible, it should be more concise.

  • Response: Thank you for your kind comment. We checked our manuscript again and modified the manuscript to fit the 'case report' format. We have removed table 3. From the manuscript and included it in the Supplementary table. Additionally, we have reviewed the content and made revisions, deleting paragraphs and sentences that were deemed unnecessary.

Reviewer 4 Report

thanks for your clarifications

Author Response

Dear Reviewer 4,

We did our best to reflect your opinions as much as possible and to raise our paper quality to suitable for publishing at Healthcare. And we are very delighted an grateful for your kind comments. We appreciate again your further comments and have striven to incorporate them in our revised manuscript again. In the following sections, please find our responses to each of your comments and suggestions.

Reviewer comment: thanks for your clarifications

  • Response: Thank you sincerely for understanding.